# *trans*-α-Necrodyl Acetate: Minor Sex Pheromone Component of the Invasive Mealybug *Delottococcus aberiae* (De Lotto)

**DOI:** 10.3390/insects16030318

**Published:** 2025-03-19

**Authors:** Javier Marzo Bargues, Sandra Vacas, Ismael Navarro Fuertes, Daniel López-Puertollano, Jaime Primo, Antonio Abad-Somovilla, Vicente Navarro-Llopis

**Affiliations:** 1Ecología y Protección Agrícola SL, Pol. Ind. Ciutat de Carlet, 46240 Carlet, Valencia, Spain; javimar@epa-ecologia.com; 2Departamento de Química Orgánica, Universitat de València, Dr Moliner 50, 46100 Burjassot, Valencia, Spain; daniel.lopez@uv.es (D.L.-P.); antonio.abad@uv.es (A.A.-S.); 3CEQA-Instituto Agroforestal del Mediterráneo, Universitat Politècnica de València, Camino de Vera s/n, edificio 6C-5ª planta, 46022 Valencia, Valencia, Spain; jprimo@ceqa.upv.es (J.P.); vinallo@ceqa.upv.es (V.N.-L.)

**Keywords:** semiochemical, insect attractant, mealybug, necrodane, citrus, integrated pest management

## Abstract

Many insect species of the superfamily Coccoidea rely on sex pheromones to find their mates and increase their populations. Their pheromone blends are generally composed of irregular monoterpenes, with one or two components. Once known, these substances could be employed to develop control methods to manage these pests and reduce their effects. Recent research has reported the main compound of the sex pheromone of the invasive mealybug *Delottococcus aberiae*, a seriously damaging pest of citrus in eastern Spain that threatens the Mediterranean area. In the present work, we reinvestigated the composition of the *D. aberiae* virgin female effluvia by volatile collection, finding an additional minor compound, *trans*-α-necrodyl acetate. Interestingly, this is not a novel compound; it has been reported as a component of the essential oil of the plant *Lavandula stoechas* subsp. *luisieri*. This compound showed male attractant activity by itself; therefore, it may participate in the communication of this species and could be a valuable tool for its control.

## 1. Introduction

Native to sub-Saharan Africa, *Delottococcus aberiae* (De Lotto) (Hemiptera: Pseudococcidae) is a pest that has recently been introduced in citrus crops of the Mediterranean area. It was first detected in 2009 in the municipality of Benifairó de les Valls (Valencia), causing considerable deformations in sweet orange and clementine fruits that entail their complete commercial depreciation [1]. Although it was described to feed on tropical as well as subtropical and temperate crops, such as coffee, guava, or olive [2], it had never been reported before as damaging for citrus fruits. Like the rest of pseudococcids, it feeds on plant sap and produces honeydew, which causes the proliferation of saprophytic fungi, as well as a decrease in the photosynthetic rate and a loss of plant vigor. Moreover, its presence can cause serious quarantine problems for citrus exports as it is a new pest for citrus fruits in Europe, until now restricted to Africa.

Currently, the active substances that are recommended and still authorized for their use against pseudococcids are mineral oil, maltodextrin, spirotetramat, acetamiprid, and sulfoxaflor [3] since the substances chlorpyrifos and methyl-chlorpyrifos, the most effective against *D. aberiae*, were withdrawn by the European Commission on 16 February 2020 (modification of Directive 91/414/EEC). Since then, the available tools have been scarce and of questionable effectiveness, and even spirotetramat and sulfoxaflor are next to be retired; thus, alternative methods were needed to manage this pest [4].

So far, α-, β-, and γ-necrodols (Figure 1), or their ester derivatives, have been reported as sex pheromones of a few insect species. The isolation and identification of the main sex pheromone component of *D. aberiae* were recently described by our group as ((4*R*)-4,5,5-trimethyl-3-methylenecyclopent-1-en-1-yl)methyl acetate ((*R*)-**1**) [5], which has proven effective in attracting the conspecific males. This substance was a new monoterpenoid with a β-necrodol skeleton to be added to the list of mealybug sex pheromones with irregular monoterpene structures [6]. Some mealybugs and other coccoid species possess multicomponent sex pheromone blends, although in many cases, a single compound provides strong biological activity. The already reported pheromone compound (*R*)-**1** demonstrated strong attractant activity for *D. aberiae* male attraction [7]. However, routine work with larger volumes of volatiles emitted by virgin females of the species allowed us to detect another component that was not present in the volatiles collected from mated females and was therefore suspected of having biological activity. In this research, the identification, characterization, and study of the biological activity of this compound, both alone and in combination with the already reported major pheromone component (*R*)-**1**, were the objectives.

## 2. Materials and Methods

### 2.1. D. aberiae Stock Colony

Mealybugs were reared on pumpkins (*Cucurbita maxima* Duchesne) to establish the main stock colony in our facilities at the Universitat Politècnica de València (UPV, Valencia, Spain). Insects were maintained in a rearing chamber, under darkness conditions, at 23 ± 2 °C, with 40–60% relative humidity. Ovisacs from the main colony were gently transferred to organic green lemons to obtain individuals for volatile collection. The lemons were previously covered with paraffin wax around the mid-section to delay their desiccation and prolong their useful life. Newly hatched individuals were established on the surface of lemons and followed their biological cycle. Groups of lemons were visually inspected every 2–3 days for the presence of male cocoons, which were manually removed with an entomological needle to leave lemons infested only with virgin females for volatile collection and profiling purposes. Other groups of lemons were left undisturbed to sample mated females. Lemons with virgin females were maintained in separate rooms under the same climate conditions. Virgin adult females were recognized from nymphs by their bigger size, more abundant cereous secretions, and the definition of their lateral wax filaments.

### 2.2. Insect Volatile Collection

Groups of five to six lemons infested with approx. 500 *D. aberiae* females (2–25 day-old virgin or mated separately) were placed in 5 L glass containers (25 cm high × 17.5 cm diameter flask) with a 10 cm open mouth. The cover had a 29/32 neck on top to fit the head of a gas washing bottle to connect downstream to a glass cartridge to trap volatiles in 3 g Porapak-Q (Supelco Inc., Torrance, CA, USA) adsorbent resin. Samples were collected continuously for 7–8 days by using an ultra-purified air stream, provided by an air compressor (Jun-air Intl. A/S, Norresundby, Denmark) coupled with an AZ 2020 air purifier system (Claind Srl, Lenno, Italy) to provide ultrapure air (amount of total hydrocarbons < 0.1 ppm). ELL-FLOW digital flowmeters (Bronkhorst High-Tech BV, Ruurlo, the Netherlands) were fitted in front of each glass container to provide an air push flow of 300 mL/min during sampling. Trapped volatiles were eluted with 20 mL pentane (Chromasolv^®^, Sigma-Aldrich, Madrid, Spain), and the resulting extracts were concentrated up to 500 µL under a gentle nitrogen stream before chromatographic analysis.

All the resulting pentane extracts were analyzed using GC–MS in a Clarus 690 GC apparatus coupled with a TSQ mass spectrometry detector (PerkinElmer Inc., Waltham, MA, USA). The GC was equipped with a ZB-5MS fused silica capillary column (30 m × 0.25 mm i.d. × 0.25 μm; Phenomenex Inc., Torrance, CA, USA). The oven was held at 40 °C for 2 min and then programmed at 5 °C/min to 180 °C before being raised to 280 °C at 10 °C/min and maintained at 280 °C for 1 min. Helium was used as carrier gas with a flow of 1 mL/min. Detection was performed in the electron impact (EI) mode (70 eV) with the ionization source set at 200 °C. Spectrum acquisition was carried out in full scan mode (*m*/*z* 35–500) and chromatograms and spectra were recorded using GC–MS Turbomass software v. 5.4 (PerkinElmer Inc., Waltham, MA, USA).

### 2.3. Identification of the Candidate Compound from Extracts of the Volatile Collections

After comparing the GC–MS volatile profiles of the virgin and mated samples (lemons infested with virgin and mated females, respectively), chromatograms showed a secondary virgin-specific compound whose MS spectra showed a peak at *m*/*z* 196, which was assumed to be the molecular ion. A micro saponification of a sample of virgin female volatile collection was performed as follows: the pentane extract of a volatile collection sample of ca. 2000 FDE (~50 ng of the candidate compound) was hydrolyzed following related procedures [8]. The extract was dried under a gentle nitrogen stream in a 2 mL GC glass vial. The residue was treated with a 0.5 M solution of K_2_CO_3_ in methanol (150 μL) and stirred for 45 min at room temperature. After this time, water (0.5 mL) was added, and the solution was extracted with dichloromethane (0.3 mL twice). The combined organic phases were concentrated with a stream of nitrogen to a volume of ca. 50 μL before being submitted to GC–MS analysis (under the abovementioned conditions).

### 2.4. Synthesis of Racemic Trans- and Cis-α-Necrodyl Acetates: (±)-(trans)-**2** and (±)-(cis)-**2**

See Appendix A for detailed experimental procedures (General Procedures, Appendix A and Appendix A).

### 2.5. Isolation of (1R, 4R)-α-Necrodyl Acetate [(1R, 4R)-**2**] from Lavandula stoechas subp. Luisieri Essential Oil

Enantiomeric pure (1*R*, 4*R*)-**2** was isolated from extracts of *Lavandula stoechas* subsp. *luisieri* specimens grown in greenhouses at the facilities of the Universitat Politècnica de València, starting from seeds of the species purchased from Cantueso Natural Seeds SL (Villarubia, Córdoba, Spain). A hundred grams of fresh plant material (stems, leaves, and flowers) were cut and extracted with 500 mL toluene by using a Soxhlet apparatus for 12 h. The solvent was selected according to its polarity and extraction efficiency, among other organic solvents. The toluene extract was concentrated in a rotary evaporator and the obtained residue (4.3 g) was subjected to chromatography with a gravity column (60 × 3.5 cm) filled with 80 g silica gel and using hexane:Et_2_O (99:1) as an eluent. The obtained fractions were analyzed using gas chromatography using the same GC–MS and conditions described before. Fractions where the presence of compound (1*R*, 4*R*)-**2** was identified in a concentration greater than 90%, as measured by the integration of the peak areas, were gathered and concentrated to afford 250 mg of the crude material. See Appendix A for details on the purification of (1*R*, 4*R*)-**2** (Appendix A).

### 2.6. Laboratory and Field Activity Bioassays

The response of *D. aberiae* males to a synthetic sample of (±)-(*trans*)-**2** was evaluated using an activity assay in a glass Petri dish. The tests were carried out with light under the same climate conditions as those of the rearing chamber at 23 ± 2 °C and 40–60% relative humidity. For these tests, males from the stock colony were collected and separated in Petri dishes just at the beginning of the formation of the cottony cocoon. After pupating and finally emerging from the cocoon, every male was observed under a stereomicroscope to check their fitness (intact legs and antennae and ability to walk normally) before being employed for the assays. During each test, a sample of (±)-(*trans*)-**2** and a negative control (pentane) were placed at opposite ends of a 30 cm diameter glass Petri dish, following described protocols [9,10,11]. One cm^2^ filter paper pieces were treated with 5 μL of the samples (ca. 100 ng of (±)-(*trans*)-**2** and pentane). Immediately, groups of five males were carefully deposited with the help of a very fine brush at the center of the dish. Then, the behavior of individuals towards stimulus sources was observed and registered for 10 min. A positive response was considered when males reached the filter paper treated with (±)-(*trans*)-**2**, whereas a negative response was those reaching the pentane source. All the males tested made a choice and, once they reached one of the filter papers, they remained there for the rest of the testing period and did not visit the other stimulus. After each test, the insects were discarded, in such a way that each male was exposed to olfactory stimuli only once. The position of the stimuli was rotated in each test. The data obtained were analyzed using the Chi-square test (χ^2^ test, *p* < 0.05).

The field activity of the different substances was evaluated under field conditions in a citrus orchard (var. Clemenules), located in the municipality of Villarreal (Castellón, Spain). The conditions of the trials carried out are summarized in Table 1. The substances to be tested were emitted from rubber septa (Ecología y Protección Agrícola SL, Carlet, Spain) dispensers, which were loaded with 100 µg of each substance by impregnation with the corresponding hexane solutions. The traps employed were 95 × 150 mm white sticky boards (Ecología y Protección Agrícola SL, Valencia, Spain). All trials used a completely randomized block design and included blank traps (only with the solvent employed for rubber septa impregnation). Within each block, traps were hung at a height of 1.5 m and were spaced 10 m apart, with each block at least 30 m apart. The traps were revised fortnightly and the number of captured males was counted under a stereomicroscope (Stemi 508; Zeiss, Oberkochen, Germany) at 50× magnification. *Delottococcus aberiae* males were recognized from other mealybug species potentially present in the orchard, e.g., *Planococcus citri* Risso, based on the wings and anal cerci, which are not visible in *D. aberiae* individuals and are obscured in those of *P. citri*.

### 2.7. Statistical Analysis

The number of captured males with each substance was compared by using generalized linear mixed models (GLMM) using R version 4.0.3 (the R Foundation for Statistical Computing 2020). For this purpose, the *glmer* function from the lme4 package was employed by assuming the Poisson error distribution. Models were constructed with the fortnightly captures as the dependent variable; substance, time (week of the study period), and their interaction (substance × time) as fixed factors; and block (experimental replicate) as random factor. The significance of the different effects was assessed by removing the corresponding factor from each model and comparing models with likelihood ratio tests. The *glht* function in the multcomp package was then used to perform Tukey HSD tests for post hoc pairwise comparisons (*p* < 0.05).

## 3. Results

### 3.1. Analysis and Elucidation of the Structure of the Minor Sex Pheromone Component of D. aberiae

The chromatographic volatile profile of lemon samples infested with either mated (Figure 2A) or virgin (Figure 2B) *D. aberiae* females revealed the presence of (R)-**1** at 20.53 min, the major sex pheromone compound first reported in Vacas et al. [11] and a minor virgin-specific peak at 19.72 min. In its mass spectrum, the base peak at *m*/*z* 121 and main fragments at *m*/*z* 134, 107, 91, and 43, together with a loss of 60 amu to give *m*/*z* 134, suggested a very close structure to **1**, an acetate ester of a necrodane-type monoterpenoid, but with one less unsaturation and with a molecular formula of C_12_H_20_O_2_. Hydrolytic basic treatment of a sample of volatile collection and subsequent analysis of the crude obtained revealed the presence of a new compound with a molecular ion of *m*/*z* 152, and main fragments at *m*/*z* 154, 139, 123, 121, and 91 (Appendix A), confirming the existence of an acetate moiety in the former.

Moreover, a comparison of the mass fragmentation with the literature of related necrodane monoterpenes led us to tentatively identify the new minor virgin-specific peak as α-necrodyl acetate, with a relative arrangement of the substituents at C-1 and C-4 that could be *trans* or *cis* (Figure 2C and Figure 2D, respectively). Both diastereomeric α-necrodyl acetates are known since they have been previously identified in other insect species, such as the carrion beetle [12], as well as in plant species, such as *Evolvulus alsinoides* (L.) (Solanales: Convolvulaceae) [13] and *Lavandula stoechas* subsp. *luisieri* (Rozeira) (Lamiales: Lamiaceae) [14]. A racemic synthesis of both diastereomers of α-necrodyl acetate was carried out to unequivocally identify the minor component emitted by *D. aberiae* females.

### 3.2. Synthesis of Racemic Cis- and Trans-α-Necrodyl Acetates

The preparation of *cis*-α-necrodyl acetate, (±)-(*cis*)-**2** was based on the retrosynthetic analysis depicted in Figure 3. In principle, the hydrogenation reaction of racemic enone **4**, previously used as an intermediate in the synthesis of the main pheromone component of *D. aberiae* [4], followed by methylenation and methanolysis steps, would give access to *cis*-β-necrodol [(±)-(*cis*)-**3**]. Subsequent isomerization of the exo double bond to the endocyclic position and acetylation of the hydroxyl group would provide access to (±)-(*cis*)-**2**. Regarding the synthesis of the diastereomer (±)-(*trans*)-**2**, its preparation could be approached, in principle, from the same intermediate (±)-(*cis*)-**3** after the change of configuration at the C-1 position of the cyclopentyl nucleus through oxidation of the hydroxymethyl moiety to the corresponding formyl group, epimerization of C-1, and reduction of the formyl group to regenerate the hydroxymethyl moiety.

### 3.3. Confirmation of the Structure and Absolute Stereochemistry of Trans-α-Necrodyl Acetate

The comparison of the mass spectra fragmentation observed for synthetic samples of β-necrodols [(±)-(*cis*)-**3** and (±)-(*trans*)-**3**] and γ-necrodol, with the necrodol obtained using hydrolysis of the compound collected from *D. aberiae* virgin female volatiles, allowed us to fully discard β and γ structures as candidates (Appendix A). Then, having both *cis*–*trans* isomers of α-necrodyl acetate in our hands, a comparative chromatographic study with the minor virgin-specific compound was carried out. Comparison of the GC–MS chromatograms of virgin volatiles collection of *D. aberiae* (Figure 2B) with pure synthetic samples of racemic *cis*-α and *trans*-α-necrodyl acetates, [(±)-(*cis*)-**2** and (±)-(*trans*)-**2**, respectively] (Figure 2C,D), allowed us to unequivocally identify the new component as the *trans* diastereomer.

Once the structure and relative stereochemistry of the new pheromone component were established, only the absolute stereochemistry of its two stereogenic centers remained to be determined. Due to the small amount of *trans*-α-necrodyl acetate **2** emitted by *D. aberiae* virgin females, the isolation of this compound from volatile collections of virgin females by using a preparatory GC–FID technique was not possible, not allowing us to assign the absolute stereochemistry of this component by comparison with enantiopure samples using chiral chromatographic techniques. However, taking into account that the main pheromone component of *D. aberiae* has been recently identified as (*R*)-(4,5,5-trimethyl-3-methylenecyclopent-1-en-1-yl)methyl acetate (*R*)-**1** (Figure 1) by our group [4], we hypothesized a common biosynthetic origin for both compounds identified in the volatile collection. Thus, we assumed a similar absolute stereochemistry in the common carbon C-4 present in both necrodane skeletons (Figure 1), as also occurs in other necrodane sex pheromones of related species, such as *Nipaecoccus viridis* (Newstead) [15] and *Pseudococcus maritimus* (Ehrhorn) [16]. Therefore, having identified the new compound as the *trans* diastereomer, we tentatively assume an absolute stereochemistry for this compound as (1*R*, 4*R*)-**2**.

To validate our hypothesis and test its biological activity, we decided to obtain an enantiomeric pure sample of (1*R*, 4*R*)-**2** from the essential oil of *L. stoechas* ssp. *luisieri* [17]. For this purpose, a sample of (1*R*, 4*R*)-**2** of 90% purity (as determined with GC–FID), obtained by chromatographic purification of the toluene plant extract obtained in the laboratory, was converted to the corresponding alcohol by treatment with K_2_CO_3_ in methanol (Figure 4), and the crude material was esterified with p-nitrobenzoic acid chloride. This solid p-nitrobenzoate ester **5** obtained was crystallized from cold hexane, resulting in a product with 99% purity estimated by GC–FID and ^1^H NMR spectroscopy. Transesterification of **5** with K_2_CO_3_ in methanol afforded (1*R*, 4*R*)-**6**, and its enantiomeric excess was determined as being higher than 98% with GC–FID with a chiral stationary phase column (Appendix A). Alcohol (1*R*, 4*R*)-**6** was acetylated using acetic anhydride and triethyl amine as the base to afford (1*R*, 4*R*)-trans-α-necrodyl acetate [(1*R*, 4*R*)-**2**] in 85% yield, which showed a purity of 99% as determined with GC–FID. An enantiomeric excess higher than 99% was determined for both enantiomers. All characterization data of (1*R*, 4*R*)-**2**, including the specific rotation, were in agreement with those previously described in the literature [11].

### 3.4. Laboratory and Field Activity Bioassays

Laboratory bioassays revealed the attractant activity of the synthetic sample of (±)-(*trans*)-**2**. *Delottococcus aberiae* males significantly preferred the filter paper baited with 100 ng of the test substance when it was presented against a blank filter paper loaded with pentane (χ^2^ = 12.8; *p* = 0.0003) (Figure 5).

In all the field trials, blank traps captured an average of fewer than 1 male/trap/week, whereas all the tested substances had significantly higher catches, regardless of the different population levels recorded in our field trials, which is due to the natural pest seasonality.

In trial 1, the substance employed to bait the traps had significant effects on male trap catches (χ^2^ = 1386.5; *p* < 0.0001). Time factor also had significant effects (χ^2^ = 423.9; *p* < 0.0001), but not the interaction of substance x week (χ^2^ = 11.7; *p* = 0.7626). The enantiomeric pure sample of (1*R*, 4*R*)-**2** obtained significantly higher male catches than the (±)-(*trans*)-**2** and (±)-(*cis*)-**2** racemates separately (Figure 6A). Interestingly, the *cis* racemate captured significantly fewer males than the *trans* racemate, and, when mixed together (1:1 blend *cis*:*trans*), male captures were also significantly lower than those of the *trans* racemate alone (Figure 6A). Accordingly, these results suggest that the presence of the opposite enantiomer of (1*R*, 4*R*)-**2** and the *cis* racemate diastereomers had detrimental effects on *D. aberiae* attracting efficacy.

In trial 2, the substance employed to bait the traps had significant effects on male trap catches (χ^2^ = 22,528.5; *p* < 0.0001). Time factor also had significant effects (χ^2^ = 8074.7; *p* < 0.0001), as well as the interaction of substance x week (χ^2^ = 1594.6; *p* < 0.0001). Data recorded during trial 2 (Figure 6B) confirmed the bioactivity of (1*R*, 4*R*)-**2**. Moreover, the results evidenced that the attractant activity of the racemate of the major pheromone compound (±)-**1** was significantly higher than that of (1*R*, 4*R*)-**2**, and, when mixing both substances in a 1:1 ratio, we found a significant additive effect.

## 4. Discussion

Chemical analyses of *D. aberiae* virgin female volatile profiles allowed us to detect a second minor component of their effluvia, which was identified as a monoterpenoid ester with a necrodane structure, ((1*R*, 4*R*)-3,4,5,5-tetramethylcyclopent-2-en-1-yl)methyl acetate ((1*R*, 4*R*)-**2**). Although not very common, necrodane skeletons are found in the plant kingdom, such as *cis*-α-necrodol in *E. alsinoides* [13] and *trans*-α-necrodol in *L. stoechas* subsp. *luisieri* [14]. Other necrodane representatives are found in the animal kingdom, such as α- and β-necrodol in defensive secretions of the carrion beetle *Necrodes surinamensis* (Fabricius) (Coleoptera: Silphidae) [12]; *trans*-α-necrodyl isobutyrate and γ-trans-necrodyl isobutyrate in the sex pheromones of the mealybug species *P. maritimus* and *N. viridis*, respectively [16,17]; and the already identified major sex pheromone compound of *D. aberiae* with a β-necrodane structure (*R*)-**1** [5,11]. In the case of *D. aberiae*, both (1*R*, 4*R*)-**2** and (*R*)-**1**, components of their effluvia share a necrodane skeleton, although with a different unsaturation degree, and probably, the structural relationship could be understood assuming a related biosynthetic pathway for both compounds. Given that the majority of sex pheromones are synthesized de novo, as they are structures not found in their host plants, it can be argued that the production of these compounds, which implies an expenditure of energy resources through the metabolism of the insect, is done exclusively out of necessity or for a specific purpose [18]. In fact, both compounds have proven biologically active to attract *D. aberiae* males by themselves. In our experiments, although (1*R*, 4*R*)-**2** displayed a weaker attractant power than (±)-**1**, it is worth noting that the 1:1 blend of (±)-**1** and (1*R*, 4*R*)-**2**) has a slightly significant additive effect. The results here presented may suggest a minor role of (1*R*, 4*R*)-**2** on *D. aberiae* male long-range attraction but it could act as a species recognition cue or as an aphrodisiac, which needs to be further investigated.

The minor component (1*R*, 4*R*)-**2** has proven attractive for *D. aberiae* males under field conditions, and the detrimental effects of using the racemic mixture (±)-(*trans*)-**2** have been observed, following the same trend reported for the main component (*R*)-**1** [5]. Not unexpectedly, the (±)-(*cis*)-**2** had a negligible attracting effect, which could possibly be attributed to traces of (±)-(*trans*)-**2** in the mixture; nevertheless, it also possesses a clear detrimental effect in the attraction as observed when baiting traps with the 1:1 blend of (±)-(*cis*)-**2**: (±)-(*trans*)-**2**, as regards the captures obtained with (±)-(*trans*)-**2** alone. The relationships between bioactivity and stereochemistry are varied in pheromone science, which has been extensively reviewed by Mori [19,20].

The attraction of insects to sex pheromones may be particularly sensitive to the composition of the pheromonal blend, which has been extensively proven for lepidopterans [21]. Of the 32 species of scale insects (Coccoidea) with reported sex pheromone blends, only seven have more than one component, with male insects being able to respond to the separate compounds, as well as to the complete blend, with different intensities [5]. For instance, the volatile profile of *Planococcus ficus* (Signoret) (Hemiptera: Pseudococcidae) showed a 5:2 blend of lavandulyl senecioate and lavandulol, with the latter showing negligible attractant activity and no synergistic effects observed on the attraction of the blend [22]. On the contrary, the pheromonal blend of *Comstockaspis perniciosa* (Comstock) (Hemiptera: Diaspididae) is composed of two regular monoterpenes in an approximately 1:1 mixture, both being bioactive compounds in male attraction, but their combination did not have synergistic effects [23]. Due to the bioactivity shown by (±)-**1** and (1*R*, 4*R*)-**2** and the additive effect of their mixture, we may hypothesize that (1*R*, 4*R*)-**2** could be a minor component of the *D. aberiae* sex pheromone complex, adding a new example to this superfamily of important agricultural pests. It is worth mentioning that the fact that both components are effective separately could generate a natural bias towards those individuals who respond better to some of the components of the pheromone complex [6]. Thus, when implementing these pheromones for pest control, it may be important to consider applying the complete pheromonal blend [24].

Semiochemicals, and specifically, sex pheromones, are highly valuable tools for pest control, especially in the current context of the restrictions imposed by the European Union directives on the use of conventional phytosanitary products (Directive 2009/128/EC of the European Parliament and of the Council on the sustainable use of pesticides). In the case of *D. aberiae*, effective control techniques are scarce and the implementation of its sex pheromone for monitoring and direct control is currently underway in eastern Spain. In this particular case, the minor component could be obtained from a natural source, the essential oil of *L. stoechas* subsp. *luisieri*, which contains more than 30% of (1*R*, 4*R*)-**2** in its composition. The biological production of insect pheromones is a growing field of study, mainly for straight-chain lepidopteran structures [25] but also recently for mealybug irregular monoterpenes [26]. The implementation of this compound to the pheromonal blend could be cost-effective, at least, for monitoring purposes or in attract-and-kill-based techniques. Identifying and developing new attractants that can be obtained from natural sources provides a clear techno–economic and environmental advantage that could be used to improve the control of *D. aberiae*. These may reduce the number of pesticides used to manage this pest in organic or integrated programs, enhance the effectiveness of the devices employed in the treatment of said species, and prevent at the same time the appearance of resistance to pheromone-based control methods.

## 5. Conclusions

This study reported the identification of (1*R*, 4*R*)-**2** as a minor sex pheromone compound emitted by *D. aberiae* virgin females. Based on the results of the bioassays, (1*R*, 4*R*)-**2** had male attractant activity by itself, although weaker than that of the main sex pheromone component (*R*)-**1**. Further trials are needed to know the pest control potential of (1*R*, 4*R*)-**2** but, taking into account the high cost of the synthesis of (*R*)-**1**, the possibility of obtaining this substance from a natural source could pose an important advantage to implement a new method for the sustainable control of this pest.

## Figures and Tables

**Figure 1 insects-16-00318-f001:**
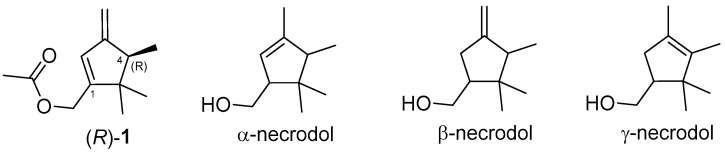
Main sex pheromone component of *D. aberiae* ((*R*)-**1**) and necrodol frameworks.

**Figure 2 insects-16-00318-f002:**
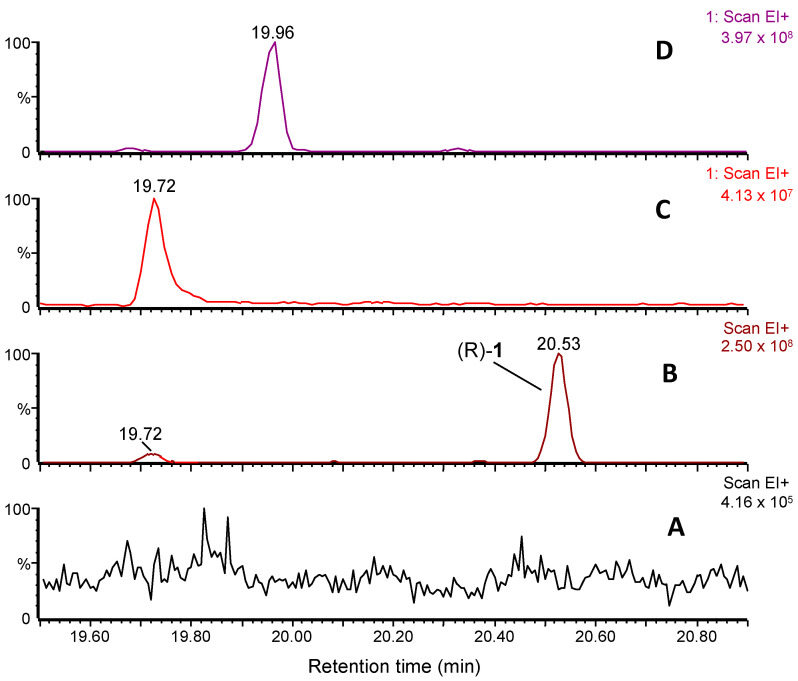
GC–MS chromatograms showing volatile collections of *D. aberiae* (**A**) mated and (**B**) virgin females. The already reported sex pheromone compound of *D. aberiae* (R)-**1** (20.53 min) and the new minor compound (19.72 min) are detected in (**B**), the latter matching the retention time of the synthetic (±)-(*trans*)-**2** (**C**) but not with (±)-(*cis*)-**2** (**D**).

**Figure 3 insects-16-00318-f003:**
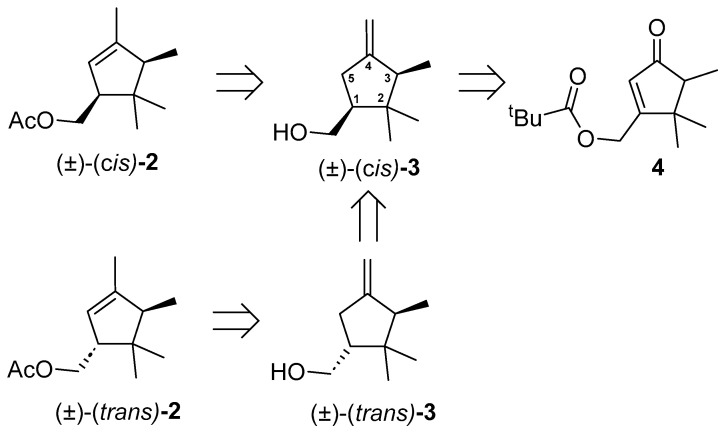
Retrosynthetic scheme for the preparation of (±)-(*cis*)-**2** and (±)-(*trans*)-**2** from cyclopentenone **4**. All compounds are racemic, for clarity, only one enantiomer is drawn.

**Figure 4 insects-16-00318-f004:**
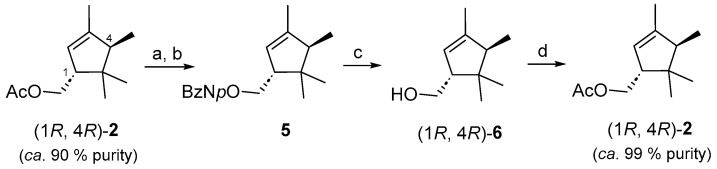
Obtainment of an enantiomeric pure sample of (1*R*, 4*R*)-trans-α-necrodyl acetate [(1*R*, 4*R*)-**2**] from *L. luisieri* essential oil extract. Reagents and conditions: (**a**) K_2_CO_3_, MeOH, r.t, 5 h, 98%; (**b**) ClCOC_6_H_4_-p-NO_2_, Et_3_N, DMAP, CH_2_Cl_2_, rt, 1 h, 90%; (**c**) K_2_CO_3_, MeOH, r.t, 24 h, 98%; (**d**) Ac_2_O, Et_3_N, DMAP, CH_2_Cl_2_, rt, 4 h, 85%.

**Figure 5 insects-16-00318-f005:**
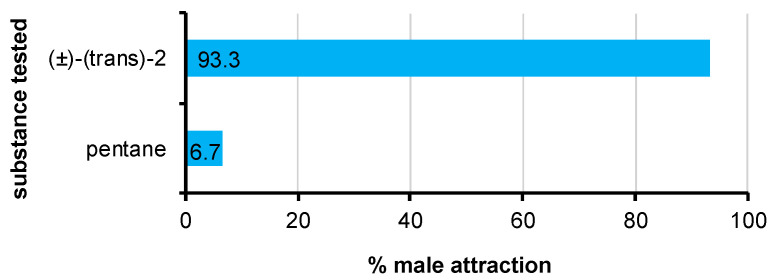
Behavioral response of male *D. aberiae* in the two-choice Petri dish laboratory bioassays: 100 ng of (±)-(*trans*)-**2** vs. solvent (pentane). Percentages were calculated according to the total number of males employed in the bioassays. Differences were significant by χ^2^ goodness of fit test (*p* < 0.001).

**Figure 6 insects-16-00318-f006:**
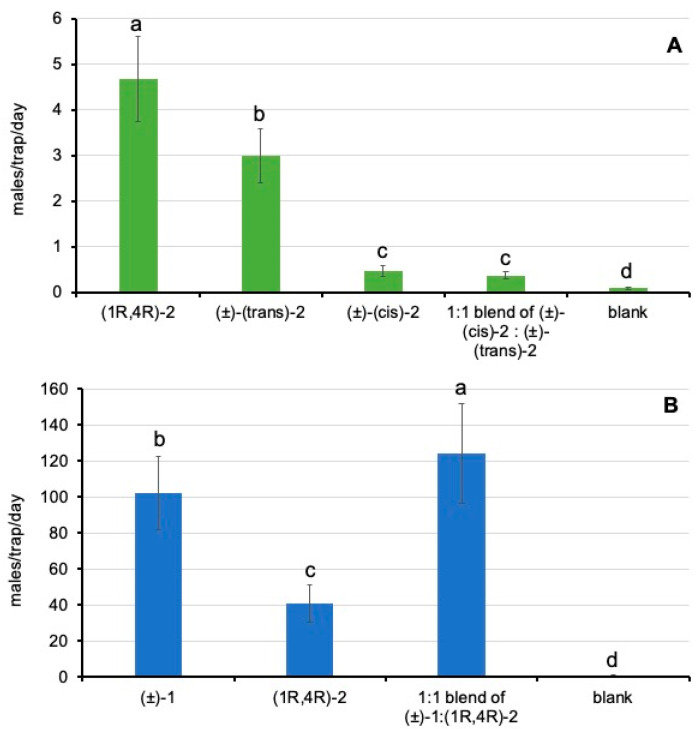
Mean (±standard error) of the number of males captured per week with the different substances tested in the field trials. (**A**) Trial 1: (1*R*, 4*R*)-**2**, (±)-(*trans*)-**2**, (±)-(*cis*)-**2**, the 1:1 blend of (±)-(*cis*)-**2**:(±)-(*trans*)-**2** and the negative control (blank); (**B**) trial 2: (1*R*, 4*R*)-**2**; (±)-**1**; 1:1 blend of (±)-**1**: (1*R*, 4*R*)-**2**] and the negative control (blank). For each trial, bars labeled with different letters differed significantly (GLMM, Tukey HSD tests, at *p* < 0.05).

**Table 1 insects-16-00318-t001:** Description of the field trials carried out to test the attractant activity of α-necrodyl acetates: (±)-(*trans*)-**2**, (±)-(*cis*)-**2**, and (1*R*, 4*R*)-**2**.

Trial	Date	#Blocks	#Weeks	Substances *
1	October 2022	4	5	Blank; (1*R*, 4*R*)-**2**; (±)-(*trans*)-**2**; (±)-(*cis*)-**2**; 1:1 blend of [(±)-(*trans*)-**2**: (±)-(*cis*)-**2**]
2	July 2023	2	4	Blank; (±)-**1**; (1*R*, 4*R*)-**2**; 1:1 blend of (1*R*, 4*R*)-**2**: (±)-**1**]

* All substances were emitted from rubber septum dispensers impregnated with the corresponding hexane solutions to provide the following loads: 100 µg (1*R*, 4*R*)-**2**; 200 µg (±)-(*trans*)-**2**; 200 µg (±)-(*cis*)-**2**; 200 µg (±)-**1**.

## Data Availability

The raw data supporting the conclusions of this article will be made available by the authors on request.

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
