# Peer review of "trans-α-Necrodyl Acetate: Minor Sex Pheromone Component of the Invasive Mealybug Delottococcus aberiae (De Lotto)"

_insects, 2025, doi:10.3390/insects16030318_

Round 1
Reviewer 1 Report
Comments and Suggestions for Authors
Dear authors,
I have reviewed your manuscript entitled trans-α-Necrodyl acetate: minor sex pheromone component of the invasive mealybug Delottococcus aberiae (De Lotto), I believe it has merit and will add to the chemical ecology knowledge of this mealybug species.
There are only a few minor questions I would like to clarify (see the pdf document attached), but, otherwise, in my opinion the paper is ready for publication.
Best regards.

Author Response
We would like to thank the revision performed by the reviewer and the consideration of our paper. Please find below a point by point list with all the comments raised and our responses. Changes made are marked in red in the revised version of the manuscript.
Comment 1: Page 3 – lines 124-125 – The saponification of a sample containing the isolated compound. The isolated compound you referring to, was it purified from the aeration extract, or from the plant extract described later (page 4)? Or is it not the isolated terpenoid here but the mixture extracted from the insects?
Response 1: The microsaponification was applied to an aeration sample (volatile collection of female effluvia) that contained the candidate compound, as you mention, it is the mixture extracted from the insects. This has been clarified as suggested (lines 124-125).
Comment 2: Page 4 – section 2.5 – Is there a particular reason for using toluene for the Spanish lavender extraction? Why not to use less toxic, lower-boiling point solvents, such hexane or petroleum ether?
Response 2: Thanks for the observation. After performing preliminary tests with several solvents such as hexane, ethyl acetate, toluene and methylene chloride, we found that toluene was slightly more efficient to extract the target compound, probably due to its intermediate polarity (included in lines 144-145). We have performed this extraction for research purposes, but we agree that it would be better to replace it for another more benign such as petroleum ether if bigger quantities of the plant material have to be extracted.
Comment 3: Page 9 – Line 319 - different letters mean the results are significantly different, don’t they?
Response 3: The reviewer is totally right, the description was mistaken. This has been amended as suggested (“bars labelled with different letter differed significantly”; line 320).
Reviewer 2 Report
Comments and Suggestions for Authors
The statement in lines 58-59 needs a reference
Please provide the morphological criteria to recognize adult females from nymphs. Also, explicit the age of individuals during aerations.
Please explain the procedure mentioned in line 158 to "check the fitness" and the criteria for selection of specimens used during the bioassays. How many replicates?
Pleas, justify the amount o chemical used in the bioassay.
Line 181: how do you recognize this species from other mealybugs present in the field, eventually, reaching the traps?
Only two replicates (blocks) were used in one field trial, why? How did you analyze the data in that case?
Line 341: why slightly? I think it´s just significant!
Author Response
We would like to thank the revision performed by the reviewer and the consideration of our paper. Please find below a point by point list with all the comments raised and our responses. Changes made are marked in red in the revised version of the manuscript.
Comment 1: The statement in lines 58-59 needs a reference.
Response 1: The reference has been included as suggested (line 59).
Comment 2: Please provide the morphological criteria to recognize adult females from nymphs. Also, explicit the age of individuals during aerations.
Response 2: Attending to not-microscopic characters easily recognizable by stereomicroscope observations, adult females are generally recognized from nymphs by their bigger size, more abundant cereous secretions and the definition of their lateral wax filaments. The size of the posterior-most pair of these filaments is commonly employed to differentiate among females of different mealybug species. A comment has been included on this regarding (lines 95-96), as well as the range of age of the virgin females employed in the volatile collection (2-25 days-old; line 97), as suggested.
Comment 3: Please explain the procedure mentioned in line 158 to "check the fitness" and the criteria for selection of specimens used during the bioassays. How many replicates?
Response 3: Every male employed for the bioassays was observed under the stereomicroscope to check that they had intact wings, antennae and they were able to walk normally (lines 159-160).
Comment 4: Please, justify the amount of chemical used in the bioassay.
Response 4: The quantity employed (100 ng) was selected according to our previous expertise and available information about Petri dish bioassays with mealybug males, where doses usually range 1-100 ng. This might be a quantity low enough to not saturate the environment and get a consistent behavioral response. References have been included (line 163).
Comment 5: Line 181: how do you recognize this species from other mealybugs present in the field, eventually, reaching the traps?
Response 5: Delottococcus aberiae was recognized from other mealybug species potentially present in the orchard, eg. Planococcus citri, according to morphological characters such as wing colour and anal cerci. Specified as suggested (lines 185-188).
Comment 6: Only two replicates (blocks) were used in one field trial, why? How did you analyze the data in that case?
Response 6: Trial 2 had 2 blocks. Each block contained one trap baited with the different treatments and we performed weekly counts for 4 weeks. Thus, we have 4 efficacy measurements for each trap. Then, we used a GLMM analysis, which can be applied to count data that not fulfil the requirements of normality and independency.
Comment 7: Line 341: why slightly? I think it´s just significant!
Response 7: We agree, results are significant in deed. Slightly has been deleted (line 347).